# Calibrated Data-Dependent Constraints with Exact Satisfaction Guarantees

**Songkai Xue**
Department of Statistics
University of Michigan
sxue@umich.edu

**Yuekai Sun**
Department of Statistics
University of Michigan
yuekai@umich.edu

**Mikhail Yurochkin**
IBM Research
MIT-IBM Watson AI Lab
mikhail.yurochkin@ibm.com

## Abstract

We consider the task of training machine learning models with data-dependent constraints. Such constraints often arise as empirical versions of expected value constraints that enforce fairness or stability goals. We reformulate data-dependent constraints so that they are *calibrated*: enforcing the reformulated constraints guarantees that their expected value counterparts are satisfied with a user-prescribed probability. The resulting optimization problem is amendable to standard stochastic optimization algorithms, and we demonstrate the efficacy of our method on a fairness-sensitive classification task where we wish to guarantee the classifier's fairness (at test time).

## 1 Motivation

In machine learning (ML) practice, accuracy is often only one of many training objectives. For example, algorithmic fairness considerations may require a credit scoring system to perform comparably on men and women. Here are a few other examples.

**Churn rate and stability** The churn rate of an ML model compared to another model is the fraction of samples on which the predictions of the two models differ [21, 30]. In ML practice, one may wish to control the churn rate between a new model and its predecessor because a high churn rate can disorient users and downstream system components. One way of training models with small churn is to enforce a churn rate constraint during training.

**Precision, recall,** *etc.* Classification and information retrieval models must often balance precision and recall. To train such models, practitioners carefully trade off one metric for the other by optimizing for one metric subject to constraints on the other.

**Resource constraints** Practitioners sometimes wish to control how often a classifier predicts a certain class due to budget or resource constraints. For example, a company that uses ML to select customers for a targeted offer may wish to constrain the fraction of customers selected for the offer. Another prominent example of a stochastic optimization problem with resource constraints is the newsvendor problem, which we come back to in section 4.

Unlike constraints on the structure of model parameters (*e.g.*, sparsity), the constraints encoding the preceding training objectives are *data-dependent*. This leads to the issue of *constraint generalization*: whether the constraints *generalize* out-of-sample. For example, if a classifier is trained to have comparable accuracy on two subpopulations in the training data, will it also have comparable accuracy on samples from the two subpopulations at test time?

36th Conference on Neural Information Processing Systems (NeurIPS 2022).

In this paper, we consider the out-of-sample generalization of *expected-value* constraints. To keep things simple, consider a stochastic optimization problem with a single *expected-value* constraint:

$$\theta^\star \in \begin{cases} \arg\min_{\theta \in \Theta} & \mathbb{E}_{P_0}\big[f(\theta; Z)\big] = \int_{\mathcal{Z}} f(\theta; z) dP_0(z) \\ \text{subject to} & \mathbb{E}_{P_0}\big[g(\theta; Z)\big] = \int_{\mathcal{Z}} g(\theta; z) dP_0(z) \leq 0 \end{cases}, \tag{1.1}$$

where $\Theta$ is a (finite-dimensional) parameter space, $f, g : \Theta \times \mathcal{Z} \to \mathbb{R}$ are (known) cost and constraint functions, and $Z \in \mathcal{Z}$ is a random variable that represents a sample. The distribution of $Z$ is unknown, so we cannot solve (1.1) directly. Instead, we obtain IID training samples $\{Z_i\}_{i=1}^n$ from the true underlying distribution $P_0$ and solve the empirical version of (1.1):

$$\widehat{\theta}_n \in \begin{cases} \arg\min_{\theta \in \Theta} & \frac{1}{n} \sum_{i=1}^n f(\theta; Z_i) \\ \text{subject to} & \frac{1}{n} \sum_{i=1}^n g(\theta; Z_i) \leq 0 \end{cases}. \tag{1.2}$$

The estimator $\widehat{\theta}_n$ (of $\theta^\star$) is guaranteed to satisfy the empirical constraint (*i.e.*, $\frac{1}{n} \sum_{i=1}^n g(\widehat{\theta}_n; Z_i) \leq 0$), but it is unclear whether $\widehat{\theta}_n$ satisfies the actual (population) constraint $\mathbb{E}_{P_0}\big[g(\theta; Z)\big] \leq 0$. As we shall see, under standard assumptions on (1.1), $\widehat{\theta}_n$ only satisfies the actual constraint with probability approaching $\frac{1}{2}$ (see corollary 2.2). This is especially problematic for constraints that encode algorithmic fairness goals. For example, the 80% rule published by the US Equal Employment Opportunity Commission, interpreted in the machine learning context, requires the rate at which a classifier predicts the advantaged label in minority groups to be at least 80% of the rate at which the classifier predicts the advantaged label in the majority group [3].

In this paper, we propose a distributionally robust version of (1.2) that *guarantees* the actual constraint $\mathbb{E}_{P_0}\big[g(\theta; Z)\big] \leq 0$ will be satisfied with probability $1 - \alpha$:

$$\widehat{\theta}_n \in \begin{cases} \arg\min_{\theta \in \Theta} & \frac{1}{n} \sum_{i=1}^n f(\theta; Z_i) \\ \text{subject to} & \sup_{P : D_\varphi(P \| \widehat{P}_n) \leq \frac{\rho_\alpha}{n}} \mathbb{E}_P\big[g(\theta; Z)\big] \leq 0 \end{cases}, \tag{1.3}$$

where $D_\varphi$ is a $\varphi$-divergence (see section 2 for details), $\widehat{P}_n$ is the empirical distribution of the training samples, and $\sqrt{\rho_\alpha}$ is the $1 - \alpha$ quantile of a standard normal random variable. More concretely, we show that $\widehat{\theta}_n$ achieves *asymptotically exact constraint satisfaction*

$$\lim_{n \to \infty} \mathbb{P}\left\{ \mathbb{E}_{P_0}\big[g(\widehat{\theta}_n; Z)\big] \leq 0 \right\} = 1 - \alpha. \tag{1.4}$$

Here the inner expectation is with respect to $Z$; the outer probability is with respect to the training samples $\{Z_i\}_{i=1}^n$. Three desirable properties of (1.3) are

1. **exact constraint satisfaction:** If the actual probability of constraint satisfaction exceeds $1 - \alpha$, then the method is too conservative. This may (unnecessarily) increase the cost of the model. By picking $\rho_\alpha$ in (1.3) carefully, constraints are satisfied with asymptotically exact probability $1 - \alpha$.
2. **computationally efficient:** As we shall see, the computational cost of solving (1.3) is comparable to the cost of solving distributionally robust sample average approximation (SAA) problems.
3. **pivotal:** There are no nuisance parameters to estimate (*e.g.*, asymptotic variances) in (1.3). The user merely needs to look up the correct quantile of the standard normal distribution for their desired level of constraint generalization.

The rest of this paper is organized as follows. In Section 2, we develop method, theory, and algorithm for stochastic optimization problems with single constraint. In Section 3, we extend our method, theory, and algorithm to stochastic optimization problems with multiple constraints. In Section 4, we validate our theory by simulating a resource-constrained newsvendor problem. In Section 5, we demonstrate the efficacy of our method by using it to train an algorithmically fair income classifier. In addition, we show how to apply our method to a fairness constrained learning problem and discuss two practical considerations for fair ML application scenarios. Finally, we summarize our work in Section 6 and point out an interesting avenue of future work.

## 1.1 Related work

The closest work to our work is [27]. They seek to pick a (data-dependent) *uncertainty set* $\mathcal{U}$ such that

$$\lim_{n \to \infty} \mathbb{P}\left\{ \sup_\theta \left\{ \mathbb{E}_{P_0}\big[g(\theta; Z)\big] - \sup_{P \in \mathcal{U}} \mathbb{E}_P\big[g(\theta; Z)\big] \right\} \leq 0 \right\} = 1 - \alpha. \tag{1.5}$$

This condition is stronger than necessary: we only require

$$\lim_{n\to\infty} \mathbb{P}\left\{\mathbb{E}_{P_0}\big[g(\widehat{\theta}_n; Z)\big] - \sup_{P\in\mathcal{U}} \mathbb{E}_P\big[g(\widehat{\theta}_n; Z)\big] \le 0\right\} = 1 - \alpha \tag{1.6}$$

where $\widehat{\theta}_n$ is a (data-dependent) estimator (not necessarily (1.2) or (1.3)). [27] study (asymptotic) constraint satisfaction (1.4) for all deterministic objective functions (see [27], §1.1 for details). They advocate picking a KL divergence ball with radius that depends on the excursion probability of a certain $\chi^2$ process.

Another closely related line of work is on data-splitting approaches for ensuring constraint generalization [37, 7]. At a high level, they split the training data into a training and validation subsets and use the validation subset to tune models trained on the training subset so that they satisfy the constraints. Although (computationally) simple and intuitive, their approach does not allow users to precisely control the constraint violation probability.

[27] is the latest in a line of work on distributionally robust optimization (DRO) that show the optimal value of DRO problems

$$\min_{\theta\in\Theta} \sup_{P\in\mathcal{U}} \mathbb{E}_P\big[g(\theta; Z)\big], \tag{1.7}$$

where $\mathcal{U}$ is a (data-dependent) uncertainty set of probability distributions, are upper confidence bounds for the optimal values of stochastic optimization problems. Common choices of uncertainty sets in DRO include uncertainty sets defined by moment or support constraints [6, 12, 22], $\varphi$-divergences [4, 26, 31], and Wasserstein distances [34, 5, 18, 28, 35]. This line of work is motivated by Owen's seminal work on empirical likelihood [32]. In recent work, [26, 15] show that the optimal value of DRO problems with empirical likelihood uncertainty sets leads to asymptotically exact upper confidence bounds for the optimal value of stochastic optimization problems ([15] consider more general $\varphi$-divergence uncertainty sets). [5] establish similar coverage results for Wasserstein uncertainty sets.

Our work is also closely related to the work on the variance regularization properties of DRO [31], which uses DRO to approximate the variance regularization cost function (see (2.4)). [20] establish similar results for Wasserstein DRO. Lastly, we relate our work to the literature on chance constrained optimization (see [24] and the references therein). The general goal of chance constrained optimization is to minimize a loss function subject to the probability of satisfying uncertain constraints is above a prescribed level. While our methods reformulate expected value constraints and we show that the solution of the reformulated problem enjoys an asymptotically exact probabilistic guarantees of constraint satisfaction. In addition, the data-dependent constraints in our work are also unknown in practice, which differs from the common setup in the chance constrained optimization literature.

## 2 Single expected value constraint

We motivate (1.3) by considering a few alternatives. First, we note that the results later in this section show that (1.2) violates the actual constraint in (1.1) approximately half the time (see corollary 2.2). The most straightforward modification of (1.2) to ensure $\widehat{\theta}_n$ satisfies the (actual) constraint $\mathbb{E}_{P_0}\big[g(\theta; Z)\big] \le 0$ is to add a "margin" in (1.3); *i.e.* enforce the constraint

$$\tfrac{1}{n}\sum_{i=1}^n g(\theta; Z_i) + \epsilon_n \le 0 \tag{2.1}$$

in (1.2). If we pick the slack term $\epsilon_n$ such that

$$\mathbb{P}\left\{\sup_{\theta\in\Theta}\left\{\mathbb{E}_{P_0}\big[g(\theta; Z)\big] - \tfrac{1}{n}\sum_{i=1}^n g(\theta; Z_i)\right\} > \epsilon_n\right\} \le \alpha,$$

then it is not hard to check that the resulting $\widehat{\theta}_n$ satisfies the (actual) constraint with probability greater than $1 - \alpha$ [36, 29]. However, this approach is most likely conservative because the constraint is unnecessarily stringent for $\theta$'s such that $\tfrac{1}{n}\sum_{i=1}^n g(\theta; Z_i)$ is less variable. It is also not pivotal: $\epsilon_n$ is often set using bounds from (uniform) concentration inequalities, which typically depend on unknown problem parameters.

To relax the empirical constraint in a way that adapts to the variability of the empirical constraints, we replace the uniform margin in (2.1) with a parameter-dependent margin:

$$\tfrac{1}{n}\sum_{i=1}^n g(\theta; Z_i) + z_\alpha \frac{\widehat{\sigma}(\theta)}{\sqrt{n}} \le 0, \tag{2.2}$$

where $z_\alpha$ is the $1 - \alpha$ quantile of a standard normal random variable and $\widehat{\sigma}^2(\theta)$ is an estimate of the asymptotic variance of $g(\theta; Z)$. We recognize the (parameter-dependent) margin as (a multiple of) the standard error of the empirical constraint. It is possible to show that enforcing (2.2) achieves asymptotically exact constraint generalization (1.4) [27].

The main issue with this method is it is not amenable to standard stochastic optimization algorithms. In particular, even if the original constraint in (1.2) is convex, (2.2) is generally non-convex. Another issue is that it is not pivotal: the user must estimate the asymptotic variance of $g(\theta; Z)$.

To overcome these two issues, we consider a distributionally robust version of (1.2); *i.e.* enforcing

$$\sup\nolimits_{P:D_\varphi(P\|\widehat{P}_n)\leq\frac{\rho_\alpha}{n}} \mathbb{E}_P\big[g(\theta; Z)\big] \leq 0, \tag{2.3}$$

where $D_\varphi(P\|Q) \triangleq \int \varphi(\frac{dP}{dQ})dQ$ is a $\varphi$-divergence. Common choices of $\varphi$ include $\varphi(t) = (t - 1)^2$ (which leads to the $\chi^2$-divergence) and $\varphi(t) = -\log t + t - 1$ (which leads to the Kullback-Leibler divergence). Although there are many other choices for the uncertainty set in (2.3), we pick an $\varphi$-divergence ball because (i) (2.3) with an $\varphi$-divergence ball is asymptotically equivalent to (2.2):

$$\sup\nolimits_{P:D_\varphi(P\|\widehat{P}_n)\leq\frac{\rho_\alpha}{n}} \mathbb{E}_P\big[g(\theta; Z)\big] \approx \frac{1}{n}\sum_{i=1}^n g(\theta; Z_i) + z_\alpha\frac{\widehat{\sigma}(\theta)}{\sqrt{n}}, \tag{2.4}$$

and (ii) it leads to pivotal uncertainty sets. For theoretical analysis, we always use $\varphi(t) = (t - 1)^2$ and $\chi^2$-divergence in the remainder of this paper.

Before we state the asymptotically exact constraint satisfaction property of (1.3) rigorously, we describe our assumptions on the problem.

1. **smoothness and concentration:** $f$ and $g$ are twice continuously differentiable with respect to $\theta$, and $f(\theta^\star; Z), \nabla f(\theta^\star; Z), g(\theta^\star; Z), \nabla g(\theta^\star; Z)$ are sub-Gaussian random variables.
2. **uniqueness:** the stochastic optimization problem with a single expected value constraint (1.1) has a unique optimal primal-dual pair $(\theta^\star, \lambda^\star)$, and $\theta^\star$ belongs to the interior of the compact set $\Theta$.
3. **strict complementarity:** $\lambda^\star > 0$.
4. **positive definiteness:** The Hessian of the Lagrangian evaluated at $(\theta^\star, \lambda^\star)$ is positive definite.

The preceding assumptions are not the most general, but they are easy to interpret. The smoothness conditions on $f$ and $g$ with respect to $\theta$, the concentration conditions of $f(\theta^\star; Z)$ and $g(\theta^\star; Z)$, and the uniqueness condition facilitate the use of standard tools from asymptotic statistics to study the large sample properties of the constraint value. The strict complementarity condition rules out problems in which the constraint is extraneous; *i.e.* problems in which the unconstrained minimum coincides with the constrained minimum.

We are ready to state the asymptotically exact constraint satisfaction property of (1.3) rigorously. The main technical result characterizes the limiting distribution of the constraint value.

**Theorem 2.1.** *Let $\widehat{\theta}_n$ be an optimal solution of (1.3) converging in probability as $n \to \infty$ to $\theta^\star$. Under the standing assumptions, we have*

$$\sqrt{n}\left(\mathbb{E}_{P_0}\big[g(\widehat{\theta}_n; Z)\big] - \underline{\mathbb{E}_{P_0}\big[g(\theta^\star; Z)\big]}\right) \xrightarrow{d} \mathcal{N}\left(-\sqrt{\rho_\alpha \operatorname{Var}_{P_0}[g(\theta^\star; Z)]}, \operatorname{Var}_{P_0}[g(\theta^\star; Z)]\right).$$

We translate this result on the constraint value to a result on constraint generalization.

**Corollary 2.2.** *Let $\sqrt{\rho_\alpha}$ be the $1 - \alpha$ quantile of a standard normal random variable. Under the conditions of theorem 2.1, we have*

$$\lim_{n\to\infty} \mathbb{P}\left\{\mathbb{E}_{P_0}\big[g(\widehat{\theta}_n; Z)\big] \leq 0\right\} = \mathbb{P}\{U \leq \sqrt{\rho_\alpha}\} = 1 - \alpha,$$

*where $U \sim \mathcal{N}(0, 1)$ is a standard Gaussian random variable.*

From theorem 2.1 and corollary 2.2 (see proofs in Appendix A), we find that

1. picking $\rho_\alpha = 0$ (*i.e.*, equivalently solving (1.2)) leads to a constraint violation probability that approaches $\frac{1}{2}$ in the large sample limit.
2. the relation between the mean and variance of the limiting distribution of the constraint value in Theorem 2.1 allows us to pick $\rho_\alpha$ in a pivotal way (*i.e.* does not depend on nuisance parameters).

## 2.1 Stochastic approximation for (1.3)

In the rest of this section, we derive a stochastic optimization algorithm to solve (1.3) efficiently. As we shall see, the computational cost of this algorithm is comparable to the cost of solving a DRO problem. The key insight is that the robust constraint function has a dual form (see Appendix J):

$$\sup_{P:D_\varphi(P\|\widehat{P}_n)\leq\rho} \mathbb{E}_P\big[g(\theta;Z)\big] = \inf_{\mu\geq 0,\nu\in\mathbb{R}} \left\{ \tfrac{1}{n}\sum_{i=1}^n \mu\varphi^*\big(\tfrac{g(\theta;Z_i)-\nu}{\mu}\big) + \mu\rho + \nu \right\}, \qquad (2.5)$$

where $\varphi^*(s) \triangleq \sup_t\{st - \varphi(t)\}$ is the convex conjugate of $\varphi$. As we use $\chi^2$-squared divergence and $\varphi(t) = (t-1)^2$, the corresponding $\varphi^*(s) = \frac{s^2}{4} + s$. The Lagrangian of (1.3) is

$$L(\theta,\lambda) \triangleq \tfrac{1}{n}\sum_{i=1}^n f(\theta;Z_i) + \lambda\sup_{P:D_\varphi(P\|\widehat{P}_n)\leq\frac{\rho_\alpha}{n}} \mathbb{E}_P\big[g(\theta;Z)\big]$$

$$= \tfrac{1}{n}\sum_{i=1}^n f(\theta;Z_i) + \lambda\inf_{\mu\geq 0,\nu\in\mathbb{R}} \left\{ \tfrac{1}{n}\sum_{i=1}^n \mu\varphi^*\big(\tfrac{g(\theta;Z_i)-\nu}{\mu}\big) + \mu\tfrac{\rho_\alpha}{n} + \nu \right\}.$$

We see that evaluating the dual function $\inf_\theta L(\theta,\lambda)$ (at a fixed $\lambda$) entails solving a stochastic optimization problem that is suitable for stochastic approximation. This suggests a dual ascent algorithm for solving (1.3):

1. evaluate the dual function at $\lambda_t$ by solving a stochastic optimization problem.
2. update $\lambda_t$ with a dual ascent step.

We summarize this algorithm in Algorithm 1. The main cost of Algorithm 1 is incurred in the third line: evaluating the dual function. Fortunately, this step is suitable for stochastic approximation, so we can leverage recent advances in the literature to reduce the (computational) cost of this step. The total cost of this algorithm is comparable to that of distributionally robust optimization.

---

**Algorithm 1** Dual ascent algorithm for (1.3)

---

1: **Input:** starting dual iterate $\lambda_0 \geq 0$
2: **repeat**
3:     Evaluate dual function:

$$(\theta_t,\mu_t,\nu_t) \leftarrow \arg\min_{\theta,\mu\geq 0,\nu} \tfrac{1}{n}\sum_{i=1}^n f(\theta;Z_i) + \lambda_t\left\{ \tfrac{1}{n}\sum_{i=1}^n \mu\varphi^*\big(\tfrac{g(\theta;Z_i)-\nu}{\mu}\big) + \mu\tfrac{\rho_\alpha}{n} + \nu \right\}$$

4:     Dual ascent update: $\lambda_{t+1} \leftarrow \left[ \lambda_t + \eta_t\left\{ \tfrac{1}{n}\sum_{i=1}^n \mu_t\varphi^*\big(\tfrac{g(\theta_t;Z_i)-\nu_t}{\mu_t}\big) + \mu_t\tfrac{\rho_\alpha}{n} + \nu_t \right\}\right]_+$
5: **until** converged

---

## 3 Multiple expected value constraints

In this section, we extend the results from the preceding section to stochastic optimization problems with multiple data-dependent constraints. Consider a stochastic optimization problem with $K$ expected value constraints

$$\theta^\star \in \begin{cases} \arg\min_{\theta\in\Theta} & \mathbb{E}_{P_0}\big[f(\theta;Z)\big] \\ \text{subject to} & \big\{\mathbb{E}_{P_0}\big[g_k(\theta;Z)\big] \leq 0\big\}_{k=1}^K \end{cases}, \qquad (3.1)$$

Following the development in Section 2, we enforce the expected value constraints with robust versions of the sample average constraints:

$$\widehat{\theta}_n \in \begin{cases} \arg\min_{\theta\in\Theta} & \tfrac{1}{n}\sum_{i=1}^n f(\theta;Z_i) \\ \text{subject to} & \big\{\sup_{P:D_\varphi(P\|\widehat{P}_n)\leq\frac{\rho_k}{n}} \mathbb{E}_P\big[g_k(\theta;Z)\big] \leq 0\big\}_{k=1}^K \end{cases}, \qquad (3.2)$$

where $\boldsymbol{\rho} = (\rho_1,\ldots,\rho_K)^\top$ are uncertainty set radii for the constraints. There are other approaches to enforcing multiple constraints that result in constraint generalization; we focus on (3.2) here because it allows the user to adjust the constraint generalization probability for different constraints.

First, we extend theorem 2.1 and corollary 2.2 to problems with multiple (expected value) constraints. We assume

1. **smoothness and concentration:** for $k \in [K]$, $f, g_k$ are twice continuously differentiable with respect to $\theta$, and $f(\theta^\star; Z), \nabla f(\theta^\star; Z), g_k(\theta^\star; Z), \nabla g_k(\theta^\star; Z)$ are sub-Gaussian random variables.
2. **uniqueness:** the stochastic optimization problem with $K$ expected value constraints (3.1) has a unique optimal primal-dual pair $(\theta^\star, \boldsymbol{\lambda}^\star)$, and $\theta^\star$ belongs to the interior of the compact set $\Theta$.
3. **strict complementarity:** $\boldsymbol{\lambda}^\star \in \mathrm{int}(\mathbb{R}_+^K)$, *i.e.*, each component of $\boldsymbol{\lambda}^\star$ is strictly positive.
4. **positive definiteness:** The Hessian of the Lagrangian evaluated at $(\theta^\star, \boldsymbol{\lambda}^\star)$ is positive definite.

The strict complementarity constraint seems especially strong here because it requires all the constraints to be active. It is possible (with extra notational overhead) to state the result in terms of just the active constraints. We refer to Section 5.1 for more information about the unknown active set. Further, as long as the sample size is large enough, the active constraints in (3.2) coincide with the active constraints in (3.1). To keep things simple, we assume all the constraints are active.

**Theorem 3.1.** *Let $\widehat{\theta}_n$ be an optimal solution of (3.2) converging in probability as $n \to \infty$ to $\theta^\star$. Under the standing assumptions, we have*

$$\sqrt{n} \begin{bmatrix} \mathbb{E}_{P_0}[g_1(\widehat{\theta}_n; Z)] \\ \vdots \\ \mathbb{E}_{P_0}[g_K(\widehat{\theta}_n; Z)] \end{bmatrix} \xrightarrow{d} \mathcal{N} \left( - \begin{bmatrix} \sqrt{\rho_1 \mathrm{Var}_{P_0}[g_1(\theta^\star; Z)]} \\ \vdots \\ \sqrt{\rho_K \mathrm{Var}_{P_0}[g_K(\theta^\star; Z)]} \end{bmatrix}, \mathrm{Var}_{P_0} \begin{bmatrix} g_1(\theta^\star; Z) \\ \vdots \\ g_K(\theta^\star; Z) \end{bmatrix} \right).$$

**Corollary 3.2.** *Under the conditions of theorem 3.1, we have*

$$\lim_{n \to \infty} \mathbb{P} \left\{ \begin{bmatrix} \mathbb{E}_{P_0}[g_1(\widehat{\theta}_n; Z)] \\ \vdots \\ \mathbb{E}_{P_0}[g_K(\widehat{\theta}_n; Z)] \end{bmatrix} \in -\mathbb{R}_+^K \right\} = \mathbb{P}\{\boldsymbol{U} \le \sqrt{\boldsymbol{\rho}}\},$$

*where $\sqrt{\boldsymbol{\rho}} = (\sqrt{\rho_1}, \ldots, \sqrt{\rho_K})^\top$, and $\boldsymbol{U}$ is a Gaussian random vector with mean zero and covariance*

$$\mathrm{Corr}_{P_0} \begin{bmatrix} g_1(\theta^\star; Z) \\ \vdots \\ g_K(\theta^\star; Z) \end{bmatrix} \triangleq D^{-\frac{1}{2}} \mathrm{Cov}_{P_0} \begin{bmatrix} g_1(\theta^\star; Z) \\ \vdots \\ g_K(\theta^\star; Z) \end{bmatrix} D^{-\frac{1}{2}}, \tag{3.3}$$

$$D \triangleq \mathrm{diag}\left(\{\mathrm{Var}_{P_0}[g_k(\theta^\star, Z)]\}_{k=1}^K\right).$$

From theorem 3.1 and corollary 3.2 (see proofs in Appendix B and C), we find that the probability of constraint satisfaction decreases *exponentially* as the number of constraints increases. We also see that our method is no longer pivotal for multiple expected value constraints: the uncertainty set radii depends on the (unknown) correlation structure among the constraint values. Fortunately, it is not hard to estimate this correlation structure. The most straightforward way is with the empirical correlation matrix. Let $\widehat{\Sigma}_n$ be the empirical covariance matrix of the constraint values. The empirical correlation matrix is then given by $\widehat{R}_n \triangleq \mathrm{diag}(\widehat{\Sigma}_n)^{-\frac{1}{2}} \widehat{\Sigma}_n \mathrm{diag}(\widehat{\Sigma}_n)^{-\frac{1}{2}}$.

Finally, it is straightforward to extend the algorithm for solving (1.3) to (3.2). The Lagrangian of (3.2) is

$$L(\theta, \boldsymbol{\lambda}) \triangleq \frac{1}{n} \sum_{i=1}^n f(\theta; Z_i) + \sum_{k=1}^K \lambda_k \sup_{P: D_\varphi(P\|P_n) \le \frac{\rho_k}{n}} \mathbb{E}_P[g_k(\theta; Z)]$$

$$= \frac{1}{n} \sum_{i=1}^n f(\theta; Z_i) + \sum_{k=1}^K \lambda_k \inf_{\mu_k \ge 0, \nu_k \in \mathbb{R}} \left\{ \frac{1}{n} \sum_{i=1}^n \mu_k \varphi^*\left(\frac{g_k(\theta; Z_i) - \nu_k}{\mu_k}\right) + \mu_k \frac{\rho_k}{n} + \nu_k \right\},$$

where we recalled the dual form of the robust constraint function (2.5) in the second step. We see that evaluating the dual function $\inf_\theta L(\theta, \boldsymbol{\lambda})$ (at a fixed $\boldsymbol{\lambda}$) entails solving a stochastic optimization problem that is suitable for stochastic approximation. This suggests a similar dual ascent algorithm for solving (1.3); we skip the details here (see Algorithm 2 in Appendix D).

# 4 Simulations

We simulate the frequency of constraint satisfaction for the following multi-item newsvendor problem:

$$\begin{aligned} \max_{\theta \in \Theta} \quad & \mathbb{E}_{P_0}[p^\top \min\{Z, \theta\} - c^\top \theta] \\ \text{subject to} \quad & \mathbb{E}_{P_0}[(\|Z^{(1)}\|_2^2 - \|\theta^{(1)}\|_2^2)_+] \le \varepsilon_1 \\ & \mathbb{E}_{P_0}[(\|Z^{(2)}\|_2^2 - \|\theta^{(2)}\|_2^2)_+] \le \varepsilon_2 \end{aligned} \tag{4.1}$$

where $c \in \mathbb{R}_+^d$ is the manufacturing cost, $p \in \mathbb{R}_+^d$ is the sell price, $\theta \in \Theta = [0, 100]^d$ is the number of items in stock, $Z \in \mathbb{R}^d$ is a random variable with probability distribution $P_0$ representing the demand, and there are $d$ items in total. The distribution $P_0$ is unknown but we observe IID samples $Z_1, \ldots, Z_n$ from $P_0$. All of the items have been partitioned into two groups so that the corresponding demand and stock can be written as $Z = (Z^{(1)}, Z^{(2)})$ and $\theta = (\theta^{(1)}, \theta^{(2)})$. The constraints in the problem exclude stock levels that underestimate the demand too much for each group of items, where $\varepsilon_1, \varepsilon_2 > 0$ indicate tolerance level of such underestimation. The target of the problem is to maximize the profit while satisfying the constraints. It is easy to rewrite the maximization problem (4.1) as a minimization problem with expected value constraints in the form of (3.1) so that we can apply our method (3.2). We pick $P_0$ as multivariate Gaussian with independent components so that the two constraints are generally uncorrelated with each other (see Appendix E for details).

Throughout the simulations, we solve (3.2) with $\boldsymbol{\rho} = (z_\alpha, z_\alpha)^\top$ for $\alpha \in \{0.4, 0.25, 0.1, 0.05, 0.005\}$. As suggested by our asymptotic theory in Section 3, the nominal probability of constraint satisfaction is $1 - \alpha$ for each constraint and $(1 - \alpha)^2$ for both constraints due to the independence setup.

In Figure 1, we plot frequencies of constraint satisfaction for each constraint and both constraints, all of which are averaged over 1000 replicates. As the sample size $n$ grows, the frequency versus probability curve converges to the theoretical dashed line of limiting probability of constraint satisfaction, validating our theory in the large sample regime. For more simulations (*e.g.*, single constraint, two dependent constraints) we refer to Appendix E.

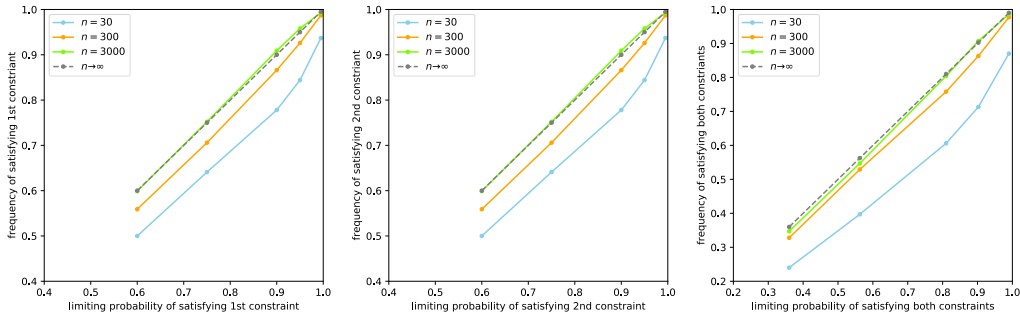

Figure 1: Frequency versus limiting probability of constraint satisfaction of the first constraint (left), the second constraint (middle), and both of the constraints (right).

## 5   Application to fair machine learning

As ML models are deployed in high-stakes decision making and decision support roles, the fairness of the models has come under increased scrutiny. In response, there is a flurry of recent work on mathematical definitions of algorithmic fairness [16, 23, 25] and algorithms to enforce the definitions [1, 10, 38].

A prominent class of fairness definitions is *group fairness*; such definitions require equality of certain metrics (*e.g.* false/true positive rates) among demographic groups. For example, consider a fair binary classification problem. Let $\mathcal{X} \subset \mathbb{R}^d$ be the input space, $\{0, 1\}$ be the set of possible labels, and $\mathcal{A}$ be the set of possible values of the protected/sensitive attribute. In this setup, training and test examples are tuples of the form $(X, A, Y) \in \mathcal{X} \times \mathcal{A} \times \mathcal{Y}$, and a classifier is a map $f : \mathcal{X} \to \{0, 1\}$. A popular definition of algorithmic fairness for binary classification is *equality of opportunity* [23].

**Definition 5.1** (equality of opportunity). *Let $Y = 1$ be the advantaged label that is associated with a positive outcome and $\widehat{Y} \triangleq f(X)$ be the output of the classifier. Equality of opportunity entails $\mathbb{P}\{\widehat{Y} = 1 \mid A = a, Y = 1\} = \mathbb{P}\{\widehat{Y} = 1 \mid A = a', Y = 1\}$ for all $a, a' \in \mathcal{A}$.*

Equality of opportunity, or true positive rate parity, means that the prediction $\widehat{Y} = h(X)$ conditioned on the advantaged label $Y = 1$ is statistically independent of the protected attribute $A$. Furthermore, an approximate version of equality of opportunity can be readily defined. We say that $\widehat{Y} = h(X)$ satisfies $\varepsilon$-*equality of opportunity* if $\mathbb{P}\{\widehat{Y} = 1 \mid A = a, Y = 1\} - \mathbb{P}\{\widehat{Y} = 1 \mid A = a', Y = 1\} \leq \varepsilon$ for for all $a, a' \in \mathcal{A}$. In this case, $\varepsilon > 0$ represents a practitioner's *tolerance* for fairness violations.

Given a parametric model space $\mathcal{H} = \{f_\theta(\cdot) : \theta \in \Theta\}$ and loss function $\ell : \Theta \times \mathcal{X} \times \mathcal{Y} \to \mathbb{R}_+$, an in-processing fair ML routine is to minimize the (empirical) risk $\mathbb{E}[\ell(\theta; X, Y)]$ while satisfying some fairness constraints. Most commonly, definitions of group fairness (including equality of opportunity, demographic parity, and more) can be written as a special example of a general set of linear constraints [1, 2] of the form $\mathbf{M}\boldsymbol{\mu}(\theta) \leq \mathbf{c}$, where matrix $\mathbf{M} \in \mathbb{R}^{K \times T}$ and vector $\mathbf{c} \in \mathbb{R}^K$ encode the constraints; $\boldsymbol{\mu}(\theta) : \Theta \to \mathbb{R}^T$ is a vector of (conditional) moments $\mu_t(\theta) = \mathbb{E}[h_t(X, A, Y, \theta) \mid \mathcal{E}_t]$ for $t \in [T]$; $g_t : \mathcal{X} \times \mathcal{A} \times \mathcal{Y} \times \Theta \to \mathbb{R}$; event $\mathcal{E}_t$ is defined with respect to $(X, A, Y)$.

This framework fits to our methodology if we note that each (conditional) moment can be written as

$$\mu_t(\theta) = \frac{\mathbb{E}_{(X,A,Y) \sim P_0}\left[h_t(X, A, Y, \theta) \times \mathbf{1}\{\mathcal{E}_t(X, Y, A)\}\right]}{\mathbb{E}_{(X,A,Y) \sim P_0}\left[\mathbf{1}\{\mathcal{E}_t(X, Y, A)\}\right]}. \tag{5.1}$$

Here the indicator $\mathbf{1}\{\mathcal{E}_t\}$ takes value 1 if the event $\mathcal{E}_t$ happens, and 0 otherwise. Moreover, we use $\mathcal{E}_t(X, A, Y)$ to emphasize that $\mathcal{E}_t$ only depends on $(X, Y, A)$ but not on $\theta$ in any way.

Note that (5.1) is a ratio of expected values, which is a non-linear statistical functional of $P_0$. To use our method, we first replace the denominator of $\mu_t(\theta)$ with an estimator, such as the unbiased estimator $\widehat{\mathbb{P}}(\mathcal{E}_t) = \frac{1}{n}\sum_{i=1}^n \mathbf{1}\{\mathcal{E}_t(X_i, A_i, Y_i)\}$. The resulting plug-in estimation of $\mu_t(\theta)$ then becomes linear in $P_0$, allowing us to apply our method (see similar tricks in [8]). We describe the application of our method to $\varepsilon$-equality of opportunity in Appendix F.

## 5.1 A two-stage method for unknown active set

In practice, it is probable that only a subset of the constraints are active. Furthermore, we do not know beforehand whether or not a constraint is active in the true population problem. To handle this scenario, we propose a two-stage method:

1. At the first stage, we solve the sample average approximation (SAA) problem (3.2) with $\boldsymbol{\rho} = \mathbf{0}_K$. By doing so, we identify the active set of the SAA problem.
2. At the second stage, we solve (3.2) with $\boldsymbol{\rho}$ such that $\rho_k$ is a positive number only if the $k$-th constraint, $k \in [K]$, was identified as active at the first stage.

In Appendix G, we show that the two-stage method also enjoys the calibration property (similar to Theorem 3.1 and Corollary 3.2) under standard assumptions (*i.e.*, strict complementarity). At a high level, the limiting probability of satisfying the true constraints depends solely on the correlation structure between active constraints and the uncertainty set radii for active constraints, as long as the SAA problem identifies active constraints with probability tending to 1.

## 5.2 Proxy dual function for non-differentiable constraints

Constraint functions in fair ML are often non-differentiable. For instance, fairness metrics are typically linear combinations of indicators that result in non-differentiable rate constraints [8–10]. This prevents the use of any gradient-based optimization algorithms. Fortunately, only the dual function evaluation step in Algorithm 1 requires access to gradients. Therefore, we can modify the algorithm by: (1) introducing proxy dual function, which uses a differentiable surrogate $\tilde{g}$ instead of the non-differentiable $g$ in the dual function evaluation step; (2) keeping $g$ in the dual ascent step. For an indicator function $h(t) = \mathbf{1}\{t > 0\}$, one can replace it by sigmoidal function $h_1(t) = (1 + e^{-at})^{-1}$ or hinge upper bound $h_2(t) = \max\{0, t + 1\}$ to produce smooth surrogates for non-differentiable rate constraints [11, 17, 9]. We summarize the proxy dual ascent algorithm in Appendix H.

## 5.3 Adult experiments

We compare the frequency of constraint satisfaction (at test time) of the sample average approximation and our methods with nominal probability $0.60, 0.75, 0.90, 0.95$ using the Adult dataset from UCI [13]. The classification task is to predict whether an individual's income per year is higher than \$50K. The fairness goal is $\varepsilon$-demographic parity ($\varepsilon$-DP): $|\mathbb{P}(\widehat{Y} = 1 \mid A = 1) - \mathbb{P}(\widehat{Y} = 1 \mid A = 0)| \leq \varepsilon$, where $A = 1$ for male is the advantaged group and $A = 0$ for female is the disadvantaged group. We use a logistic regression model for classification and techniques in this section for implementation.

In Figure 2, we have line plots for frequency of constraint satisfaction and box plots for classification error rate, all of which are summarized over 100 replicates. The left panel shows that solving (3.1)

directly leads to one half chance of constraint violation, while our method's constraint satisfaction frequency matches its nominal value. The price of a higher chance of test-time fairness satisfaction is an increase in classification error rate as shown in the right panel. From the baseline to $95\%$ chance of fairness satisfaction, we basically trade off $2\%$ increase in error rate. We refer to Appendix I and K for details and more experiments.

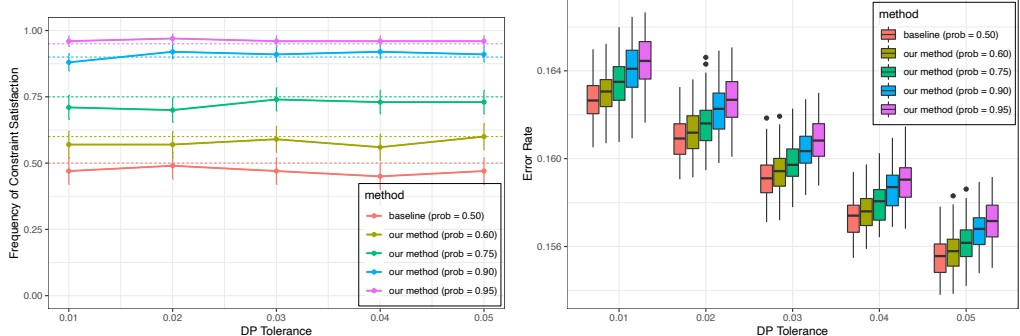

Figure 2: Frequency of constraint satisfaction (left) and classification error rate (right) for different demographic parity tolerance $\varepsilon \in \{0.01, 0.02, 0.03, 0.04, 0.05\}$. Baseline (sample average approximation, SAA) and our methods (with nominal probability $0.60, 0.75, 0.90, 0.95$) are compared.

## 6   Summary and discussion

We explore the problem of exact constraint satisfaction probability in stochastic optimization with expected-value constraints. We propose a distributionally robust reformulation of data-dependent constraints and provide a theoretical guarantee of constraint satisfaction with an asymptotically exact probability specified by the user. For solving the reformulated problem, a scalable dual ascent algorithm and its variants are proposed. The computational cost of our algorithm is comparable to that of a standard distributionally robust optimization problem. Our theory on exact constraint satisfaction probability is validated via simulations on the resource-constrained newsvendor problem. The efficacy of our methods is empirically demonstrated on fair machine learning applications.

Some data-dependent constraints are by nature *non-linear* in the underlying probability measure. For example, (5.1) is a ratio of expected values. An intriguing direction for future research is to generalize the methods and theory developed in this work to constraints on non-linear functions of expected values. Such forms of constraints are known as *statistical functionals* in statistics literature [19]. The non-linear dependence of the constraint function on the probability measure precludes the stochastic approximation as a general way of evaluating the dual function, as the constraint function no longer admits a dual form (2.5), calling for the development of a new algorithm.

## Acknowledgments and Disclosure of Funding

This paper is based upon work supported by the National Science Foundation (NSF) under grants no. 1916271, 2027737, and 2113373.

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
