# OpenReview forum: "Calibrated Data-Dependent Constraints with Exact Satisfaction Guarantees"
_NeurIPS.cc/2022/Conference — NeurIPS 2022 Accept_

### Official Review · Reviewer_oKKe · 2022-07-07

**Rating:** 7
**Confidence:** 3
**Soundness:** 3 good
**Presentation:** 3 good
**Contribution:** 3 good

**Summary:**

This paper proposes a distributionally robust optimization algorithm to machine learning problems with constraints in the form of expected values. Performing the proposed DRO with the empirical version of the constraints acts as a variance-estimation-free way of introducing margins so that the data dependent solution can achieve exact constraint satisfaction in terms of the expectations. The proposed method is compatible with standard stochastic optimization in its dual form, and empirical studies show that the method is indeed calibrated and is effective for multiple fairness sensitive classification tasks.

**Questions:**

The reviewer does not have major questions regarding the manuscript. Here are a few minor questions:
- Since the dual form of the DRO is key to running the computation, it is worth a bit more attention and explanation in Section 2.1.
- If the reviewer understands correctly, the "pivotalness" is a direct result of the chi-square divergence. It would be nice to have more discussions here.

**Limitations:**

See weaknesses.

**Strengths And Weaknesses:**

Strengths:
The paper presents a complete study on the exact constraint satisfaction perspective of ML fairness, which is a widely-recognized and important topic. The authors introduces DRO with chi-square divergence bounding the empirical and the population distributions, which can be viewed as a replacement for parameter-dependent constraint margins in other literature, resulting in an optimization problem that is both more convex and does not require the estimate the variability of constraint violation. This is a theoretically sound conclusion. The authors also have covered most alternatives or fixes for practical nuisances when applying the algorithm, which makes the method easily feasible for lots of real world questions. The presentation in the paper is clear and easy to follow.

Weaknesses:
The proposed method only guarantees "exactness" for a single constraint expressed as an expectation. Despite that the authors have introduced adaptations for multiple constraints in other forms (e.g. ratio of expectations), the "exactness" might be subtly impacted by the plug-in estimates used there.

---

> ### Author Response · Authors · 2022-08-02
> **Response to Reviewer oKKe**
>
> We thank the reviewer for the valuable feedback. We address questions and concerns below.
>
> **Despite that the authors have introduced adaptations for multiple constraints in other forms (e.g. ratio of expectations), the "exactness" might be subtly impacted by the plug-in estimates used there.**
>
> The reviewer is correct in noting that our theory cannot directly handle constraints that are ratios of expectations. That said, we show that the methods perform well empirically in Section 5.3 (i.e. lead to solutions with exact constraint satisfaction guarantees). Unfortunately, the gap in our theory is non-trivial; filling the gap entails a careful perturbation analysis of stochastic optimization problems with ratios of expectations constraints that is beyond the scope of this paper.
>
> **Since the dual form of the DRO is key to running the computation, it is worth a bit more attention and explanation in Section 2.1.**
>
> The dual form of the DRO is standard, and we provide a standard derivation of it in appendix J, line 656 - 665, of the revised manuscript. We will elaborate on it in Section 2.1 right after equation (2.5) if we have more space in the main text.
>
> **If the reviewer understands correctly, the "pivotalness" is a direct result of the chi-square divergence. It would be nice to have more discussions here.**
>
> This is an interesting question that could inspire future research. We suspect that it is possible to preserve the "pivotalness" even if the chi-squared divergence is replaced by any $\varphi$-divergence (subject to standard assumptions on $\varphi$). However, there are some technical challenges in generalizing our results to any $\varphi$-divergence. For general $\varphi$-divergence, we do not have the variance-regularization expansion (see line 487 in Appendix) exactly, and therefore it is hard to differentiate such formula with respect to the model parameter $\theta$.

---

> > ### Comment · Reviewer_oKKe · 2022-08-09
> > **response to rebuttal**
> >
> > I would like to thank the authors for the rebuttal addressing my concerns. After reading I would like to keep my initial rating of accept.

---

### Official Review · Reviewer_tHYG · 2022-07-10

**Rating:** 7
**Confidence:** 4
**Soundness:** 4 excellent
**Presentation:** 4 excellent
**Contribution:** 3 good

**Summary:**

This paper considers the problem of stochastic optimization subject to data-dependent constraints in the form of E g(\theta;Z) <= 0. Solving these problems using finite samples raises the issue of ensuring that the constraints will be satisfied on the population distribution, and not just on the samples. This paper proposes to address this issue by solving a distributionally robust variant of the problem, where the empirical loss is minimized subject to the constraint that E_P g(\theta;Z) <= 0 for all probability distributions P that are sufficiently close to the empirical distribution in the sense that the phi-divergence between them is less than a certain size. The paper shows that this allows for calibrated satisfaction of the constraints asymptotically, i.e. in the large sample limit, the constraints will be satisfied with probability exactly 1-alpha for alpha chosen by the user. Several examples are also discussed, as are a few tricks for solving the associated optimization problem.

**Questions:**

I noticed in almost all of the experiments that the frequency of satisfying the constraints seems to nearly always approach the asymptotic limit from below. This suggests to me that the distributionally robust modification of the constraint is generally over-permissive, and only becomes correctly calibrated in the limit (or at least once the CLT kicks in). Is it clear that this would be the case? If you are concerned about what happens with relatively few finite samples, do you have any suggestions about how to adjust the method to improve calibration here?

**Limitations:**

See "weaknesses" above.

**Strengths And Weaknesses:**

Strengths:
This paper is very clearly written. Each idea is introduced nicely and each step in the discussion follows very nicely from the previous one, to the extent that everything seems almost obvious in hindsight. Having the constraints' satisfaction rate be calibrated in this way seems very useful in practice and this paper presents a very straightforward recipe for achieving this asymptotically. Finally, the experiments indicate that although the theoretical results in the paper are only asymptotic, they appear to hold already at practical sample sizes.
Weaknesses:
I think it would be helpful to discuss also the computational issues that arise in trying to solve the optimization problem associated with the method. E.g., the optimization problem in line 3 of Algorithm 1 is generally non-convex (unless the constraint g(\theta;Z_i) is linear in \theta). Of course, given the current practice of machine learning, we may not actually be so concerned about this, and we might just use SGD and hope that it works. But in this case, I wonder if anything can be said about what happens when (1.3) is solved only approximately? If you solve it to some epsilon error, is it clear what happens to the satisfaction rate? This is perhaps too big of a question to fully address in this paper, which focuses on the statistical issues at play, but I think that computational issues deserve at least some discussion in a machine learning venue such as Neurips.

---

> ### Author Response · Authors · 2022-08-02
> **Response to Reviewer tHYG**
>
> We thank the reviewer for the valuable feedback. We address questions and concerns below.
>
> **... the optimization problem in line 3 of Algorithm 1 is generally non-convex ... in this case, I wonder if anything can be said about what happens when (1.3) is solved only approximately?**
>
> Our theory relies on a perturbation analysis of the KKT conditions; thus it applies to local minima in non-convex problems because local minima also satisfy the KKT conditions (as long as the corresponding local minima at the population level satisfies our assumptions). That said, for non-convex problems, there may be multiple local minima even at the population level, so one must "match" empirical local minima with their population counterparts before appealing to our theoretical results.
>
> **If you solve it to some epsilon error, is it clear what happens to the satisfaction rate?**
>
> This is a computational issue to be worth discussing. Assume that (1.3) can be solved to $\epsilon$ error for any small positive number $\epsilon$. As long as $\epsilon$ is of order little-o of 1 over root-$n$ (here $n$ is the sample size), the approximate solution has the same statistical property as the true solution $\hat{\theta}_n$ (for example, the $\hat{\theta}_n$ in Theorem 2.1 and 3.1). In other words, $o(1/\sqrt{n})$ is an acceptable amount of optimization error for an estimator to maintain its statistical properties. If the optimization error is larger than $o(1/\sqrt{n})$, then the optimization error will affect the statistical properties of the computed solution. This type of results is frequently discussed in classic statistical literature regarding M-estimators and Z-estimators.
>
> **I noticed in almost all of the experiments that the frequency of satisfying the constraints seems to nearly always approach the asymptotic limit from below. This suggests to me that the distributionally robust modification of the constraint is generally over-permissive, and only becomes correctly calibrated in the limit (or at least once the CLT kicks in). Is it clear that this would be the case?**
>
> It is not clear that the distributionally robust version of the constraint is always conservative; the finite-sample properties of the distributionally robust version of the constraint is problem-dependent. That said, it is not surprising that the distributionally robust version of the constraint is often conservative because the statistical properties of the distributionally robust version of the constraint is similar to that of a sample mean and Z-intervals centered at sample means are often conservative.
>
> **If you are concerned about what happens with relatively few finite samples, do you have any suggestions about how to adjust the method to improve calibration here?**
>
> This question can be rephrased as follows: is it possible to more precisely approximate the probability of constraint satisfaction in the finite sample regime? The theory in this work builds on the first-order expansion of the DRO formula (see equation (2.4), also referred to as the variance regularization property of divergence-based DRO). This leads to a tractable and scalable algorithm with nice pivotal property. Technically speaking, one can consider higher-order expansions of equation (2.4). This could lead to higher-order corrections of the distributionally robust version of the constraint.

---

> > ### Comment · Reviewer_tHYG · 2022-08-04
> > **thanks**
> >
> > Thanks for the response, nice paper!

---

### Official Review · Reviewer_EQuq · 2022-07-14

**Rating:** 7
**Confidence:** 3
**Soundness:** 3 good
**Presentation:** 3 good
**Contribution:** 3 good

**Summary:**

This paper addresses the problem of constraints satisfaction in Machine Learning. More precisely, the goal is to guarantee that, with high probability, a model learned to satisfy an empirical estimate of a given constraint will also satisfy the constraint with respect to the original distribution. Based on recent advances in Distributionally Robust Optimization, the paper proposes a new approach to learn models that are theoretically guaranteed to exhibit such behaviour. Empirically, the interest of the proposed algorithm is demonstrated in an application to Fair Machine Learning.

**Questions:**

My main concerns with this paper are detailed in comments 4., 5., and 6. in the Strengths and Weaknesses section and would need to be addressed.

**Strengths And Weaknesses:**

Pros:
 - The paper is well written and relatively easy to follow.
 - The problem of exact constraints satisfaction is important.
 - The proposed approach is theoretically well founded.

Cons:
 - The proposed algorithm does not come with guarantees on the quality of its solutions.
 - Group fairness constraints do not satisfy the main assumptions in the theoretical analysis.
 - There is a lack of datasets diversity and baselines in the experiments.

On the one hand, this paper addresses the important problem of exact constraint satisfaction in machine learning in a theoretically sound way. On the other hand, there is a gap between the developed theory and the main motivating example, that is fairness. Overall, the merits of this paper outweight its flaws and it could be accepted.

Detailed comments:
1. This paper is well written and relatively easy to follow. The assumptions on the problem are necessary for the theoretical derivations are explicitly stated and intuition is provided on the meaning of the various results.

2. The problem addressed in the paper is significant. Indeed, models learned on a limited training set encounter new examples when they are deployed in the wild and ensuring that the desired constraints are respected on these new examples (as long as they follow the distribution that generated the training samples) is important. This is particularly true for fairness constraints which is one of the motivating problems of this paper.

3. To ensure that the learned model satisfies a constraint exactly, that is even on the overall unknown distribution, the paper proposes to ensure that the empirical estimate is satisfied with a sufficiently large margin. It is then shown that it is similar to solving a distributionally robust optimization problem where the constraint has to be satisfied for any distribution that lies in a ball around the empirical distribution of the training sample. It is then theoretically shown that, in the large sample limit, the solution to this optimization problem satisfies the exact constraint provided that the radius of the ball considered is sufficient large and that the constraint under consideration respect some constraints. Similar results are provided in the case where several constraints should be satisfied.

4. To solve the distributionally robust optimization problem, the paper proposes to use a stochastic scheme based on Lagrange multipliers. Due to its nature, this approach is likely to lead to approximate solutions and it is not clear as to how good these solutions will be compared to the optimal one. Similarly, in Section 5, several heuristic are proposed to deal with the problems of unknown active sets and non-differentiable constraints but their impact on the quality of the solutions is not discussed.

5. One of the motivating problems for the proposed approach is group fairness constraints in machine learning as detailed in the experimental Section 5. Unfortunately, there seems to be a gap between the theory and algorithm developed in the paper and this setting:
    - Fairness constraints do not satisfy the assumptions necessary for the theory to hold. Indeed, group fairness constraints are usually non-differentiable.
    - In the case of exact fairness, the proposed approach might lead to unfeasible problems. Indeed, in this setting, the goal is, for example, to ensure that the probability of correctly predicting the desirable outcome given that it should be predicted is independent of the sensitive attribute (equality of opportunity, Definition 5.1). It corresponds to an equality constraint (where a difference should be equal to $0$) that is often broken down in two inequality constraints which can only be simultaneously satisfied with a margin of $0$. The interpretation given in Equation (2.4) shows that this is not compatible with the proposed approach.
    - The previous issue is less likely to happen with approximate fairness where the constraints have to be satisfied up to a factor $\epsilon$. However, in this case, it could be that when $\epsilon$ is smaller than the margin necessary to ensure a given probability of satisfying the exact constraints, the problem becomes infeasible again. Consequently, an interesting extension of this work would be to provide guarantees on the probability that the exact constraints will be satisfied given that the empirical constraints are satisfied up to a given margin.

6. The experiments investigate the interest of the proposed approach in two different settings.
    - In a simulated environment, it is shown that the, as the number of training examples increases, the proposed approach will indeed learn models that satisfy the exact constraints with the desired probability.
    - On a fair machine learning problem, it is shown that using the proposed approach indeed increases the probability of constraints satisfaction. Unfortunately, this result is only shown on a single dataset (Adult) and a single baseline is considered (directly solving the problem with the empirical constraints). It would have been interesting to provide more extensive experiments by, for example, considering [1], [10], and [23] as baselines and considering other datasets (the ones used in the aforementioned papers for example).

---

> ### Author Response · Authors · 2022-08-02
> **Response to Reviewer EQuq [Part 1]**
>
> We thank the reviewer for the valuable feedback. We address questions and concerns below.
>
> **To solve the distributionally robust optimization problem, the paper proposes to use a stochastic scheme based on Lagrange multipliers. Due to its nature, this approach is likely to lead to approximate solutions and it is not clear as to how good these solutions will be compared to the optimal one.**
>
> The algorithm we propose to solve the DRO problem (Algorithm 1) is essentially dual ascent, which is guaranteed to converge under standard conditions on the dual function. We also use this algorithm to solve the DRO problem in our computational results, and the computed solutions are accurate enough to reproduce the statistical properties of the exact solutions.
>
> **Similarly, in Section 5, several heuristic are proposed to deal with the problems of unknown active sets and non-differentiable constraints but their impact on the quality of the solutions is not discussed.**
>
> Regarding the two-stage procedure for handling unknown active sets, we show in Appendix G that the first stage identifies the active set with high probability (see Proposition G.1) and the second stage leads to a solution with exact constraint satisfaction guarantees (Theorem G.2).
>
> Regarding non-differentiable constraints, we have no theoretical results guaranteeing the DRO approach has exact constraint satisfaction properties when the constraints are non-differentiable, but we show empirically in Section 5.3 that solving the DRO problem with the proxy-Lagrangian approach of Cotter et al [10] leads to solutions with the desired exact constraint satisfaction properties (see Figure 2).
>
> **Fairness constraints do not satisfy the assumptions necessary for the theory to hold. Indeed, group fairness constraints are usually non-differentiable.**
>
> The reviewer is correct in noting that demographic parity and equalized odds/opportunity constraints are non-differentiable and therefore do not satisfy the theory's assumptions. However, there are group fairness constraints (e.g. risk parity constraints) that are differentiable. Although we have no theoretical results guaranteeing the DRO approach has exact constraint satisfaction properties when the constraints are non-differentiable, we show empirically in Section 5.3 that solving the DRO problem with the proxy-Lagrangian approach of Cotter et al [10] leads to solutions with the desired exact constraint satisfaction properties (see Figure 2).
>
> **In the case of exact fairness, the proposed approach might lead to unfeasible problems. ... an equality constraint (where a difference should be equal to 0) that is often broken down in two inequality constraints which can only be simultaneously satisfied with a margin of 0 ...**
>
> The reviewer is correct in pointing out that enforcing some fairness constraints exactly may lead to infeasible sample average approximation problems. In this case, our approach is not applicable. To avoid this issue, we consider approximate fairness (which leaves a positive margin) in the application section and make a strict complementarity assumption (which excludes the case where two inequality constraints can only be satisfied simultaneously with a margin of zero) in the theory section. The use of approximate fairness is prevalent in the literature; see [1-2, 8-10] for examples.
>
> **The previous issue is less likely to happen with approximate fairness where the constraints have to be satisfied up to a factor $\epsilon$. However, in this case, it could be that when $\epsilon$ is smaller than the margin necessary to ensure a given probability of satisfying the exact constraints, the problem becomes infeasible again.**
>
> The extra margin that we add to obtain exact constraint satisfaction is $O(\frac{1}{\sqrt{n}})$, so this extra margin becomes vanishingly small as the sample size $n$ grows. Thus this issue (the SAA problem is feasible but its distributionally robust counterpart is infeasible) does not arise  as long as $n$ is large enough.
>
> **Consequently, an interesting extension of this work would be to provide guarantees on the probability that the exact constraints will be satisfied given that the empirical constraints are satisfied up to a given margin.**
>
> This depends on the form of the exact constraints. If they are inequality constraints, then it is possible to leverage some of the same theoretical tools to estimate the probability that a solution to a SAA problem with extra margins will satisfy the exact constraints. If they are equality constraints (e.g. enforcing exact group fairness), then the set of solutions that satisfy the exact constraints will generally be a lower-dimensional set. In this case, the probability that a solution of a SAA problem with extra margins falls in this lower-dimensional set (i.e. satisfies the exact constraints) is generally zero.

---

> > ### Author Response · Authors · 2022-08-02
> > **Response to Reviewer EQuq [Part 2]**
> >
> > **On a fair machine learning problem, it is shown that using the proposed approach indeed increases the probability of constraints satisfaction. Unfortunately, this result is only shown on a single dataset (Adult) and a single baseline is considered (directly solving the problem with the empirical constraints). It would have been interesting to provide more extensive experiments by, for example, considering [1], [10], and [23] as baselines and considering other datasets (the ones used in the aforementioned papers for example).**
> >
> > We conduct more experiments using two-dataset approach of Cotter et al [8] as an additional baseline and UCI default of credit card clients data as an additional dataset. The new experiments are included in Appendix K (line 666 - 702) of the revised manuscript. [1] and [10] are essentially sample average approximation (SAA), which is the baseline utilized in the original manuscript. We do not include [23] as a baseline because it is a post-processing method (our method and the other baselines are in-processing methods).

---

> > > ### Comment · Reviewer_EQuq · 2022-08-09
> > > **Thanks for the additional details.**
> > >
> > > Thanks for the additional details. The rebuttal addressed my concerns and, as such, I increased my score.

---

### Meta-Review · Area_Chair_BxRi · 2022-08-31

**Recommendation:** Accept
**Confidence:** Certain

**Metareview:**

The reviewers unanimously think the paper is worth publishing. I agree with them. The reviewers did a good job addressing reviewer concerns in rebuttal as well. This paper should be accepted.

**Award:**

Yes

---

### Decision · Program_Chairs · 2022-09-14

Accept